# ACCELERATING DIFFUSION MODEL WITH DYNAMIC ALIGNMENT

## ABSTRACT

Recent studies have shown improvements in both generation quality and training efficiency by constraining representations during the denoising process of generative diffusion models. While distilling simple visual representations is effective, it can lead to over-alignment issues. When the model achieves alignment early in training, these simple representations can become hindrance to training the generative capacity. Building upon prior efforts that addressed this problem from the perspectives of alignment objectives and training strategies, we introduce DyA. First, we incorporate richer alignment materials to address the problem of overly simplistic representations at the source. Second, we use the internal denoising time of the diffusion model as an indicator variable to dynamically adjust the constraint strength of different levels of information. Finally, we employ the Stochastic Dropout Strategy (SDS), which allows the model to emphasize generative capacity training while providing guidance throughout the entire process. Experiments have shown that this approach improves both generation quality and training efficiency. The DyA accelerates SiT training by approximately 20 times, achieving performance comparable to SiT-XL model trained for 7M steps in just around 350K steps.

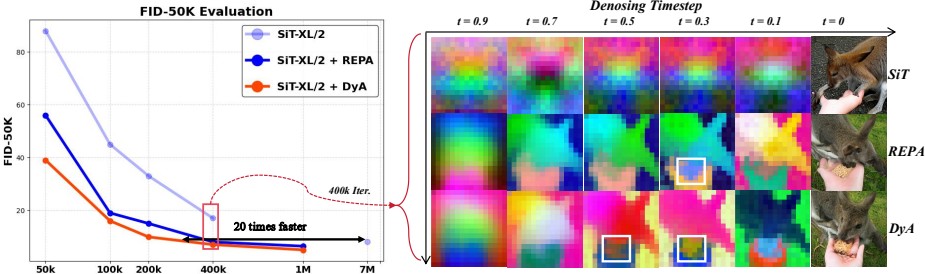

Figure 1: The **Dynamic Alignment** method has achieved further improvements compared to the REPA method, achieves convergence that is about $20.0\times$ faster than the SiT. From the visualizations of the student block representations in the target model, the features of detail emerge at earlier denoising steps and are clearly disentangled from that of the background. DyA produces significantly sharper and more accurate details than both the SiT and REPA baselines, corroborating the quantitative improvements.

## 1 INTRODUCTION

Diffusion-based generative models (Ho et al., 2020; Song et al., 2020) and flow-based models(Albergo & Vanden-Eijnden, 2023; Lipman et al., 2023; Liu et al., 2023) have significant progress in generation task(Podell et al., 2023; Saharia et al., 2022a; Esser et al., 2024b; Min et al., 2023). Recent works have also explored the use of diffusion models as representation learners and produce discriminative features(Li et al., 2023; Xiang et al., 2023; Chen et al., 2024; Mukhopadhyay et al., 2023). Methods (Leng et al., 2025; Tian et al., 2025; Yu et al., 2025) use alignment technique between the middle layer of a diffusion model and representation from a pretrained teacher model, effectively accelerating model training and enhancing the quality of generated outputs. However,

the simple representations from the teacher model may become a hindrance to the model's generative capacity training in the later stages. Seeing alignment as a variant of knowledge distillation(Fan et al., 2024; Zhao et al., 2022) makes the problem clearer. This phenomenon can be attributed to **capacity mismatch**, which arises because the representations from the teacher model are too simplistic to the complexity of the generative task of student model.

Essentially, pre-training visual encoders compress images into discriminative high-level representations; their objective rewards invariance—robustness to translation, illumination, and occlusion—while actively discarding task-irrelevant details. In contrast, diffusion models must reconstruct the original image with pixel-level fidelity, requiring them to model the full high-order statistics of the data distribution. Diffusion models retain copious high-frequency information through multi-scale skip connections, yielding "draft-like" redundant features, whereas pre-trained features behave like "summaries" that maximize inter-class separability in a low-dimensional semantic space. The divergence between the two feature types is rooted in their objectives: discriminative modeling seeks local invariance and semantic compression, whereas generative modeling seeks global fidelity and detail preservation.

Drawing on previous studies that analyzed model training from a gradient perspective (Guo et al., 2024; Wang et al., 2025), the impact of REPA loss on the model's generative capabilities varies over time. As shown in Figure 6, the gradient of auxiliary loss has almost no effect on generative capabilities at the initial training stage. As training progresses, it even have a negative impact.

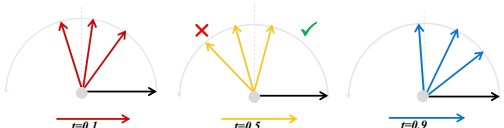

Figure 2: Black arrows indicate the gradient direction of the Denoising Loss, and colored represent that of the Auxiliary Loss. The angle between them increases as the training stage advances.

Suppose we redefine the core task as leveraging simple representation to guide complex representations, rather than a generation problem, inspired by the successes of knowledge distillation(Huang et al., 2022) and representation alignment based generation(Yu et al., 2025; Wang et al., 2025), we recast the above issue as two sub-problems under a unified generative framework. On the one hand, the complexity of the teacher representations needs to match the complexity of the generative task more closely to prevent the issue of capacity mismatch. On the other hand, the training objective of the model should emphasize the training of generative capabilities, treating the alignment of representations merely as an auxiliary tool.

We propose Dynamic Alignment (DyA) to address these problems. First, the DyA introduces multi-level information to address the issue of hindered generative capacity caused by overly simplistic alignment targets in former methods.

Additionally, the DyA implement a temporal module, which uses the denoising time of the diffusion model as an indicator variable to adjust the alignment constraints intensity of different level information, bringing greater flexibility to the training process.

Second, we introduce the Stochastic Dropout Strategy (SDS), which addresses the second sub-problem from the perspective of training methodology. In the SDS, the model halts alignment behavior in every training iteration with a pre-set probability $p$, retaining only the denoising objective tied to generative capacity in the loss term. Unlike the truncation strategy (Wang et al., 2025), which stops alignment from a pre-set step, SDS prioritizes generative capacity while continuously guiding the model throughout whole training process. Experiments show that SDS brings a greater enhancement to the model's generative quality.

Our main contributions are:

- We propose Dynamic Alignment (DyA), a multi-level guidance strategy that dynamically modulates alignment strength using diffusion timestep, mitigating representation collapse in later training stages.

- We introduce Stochastic Dropout Strategy (SDS), a probabilistic training mechanism that encourages generative capacity focus while retaining auxiliary guidance adaptively.

- Extensive experiments on ImageNet and ArtBench demonstrate $20\times$ acceleration and superior generation quality, surpassing the REPA and vanilla SiT baselines.

## 2 RELATED WORK

Recent developments in diffusion models have largely revolved around enhancements in learning methods, sampling strategies(Song et al., 2022; Song & Ermon, 2019; Lu et al., 2022; 2023), guidance techniques(Ho & Salimans, 2022; Nichol et al., 2022), latent representations(Rombach et al., 2022), and overall model structures(Ho et al., 2022; Peebles & Xie, 2023; Saharia et al., 2022b; Xue et al., 2023). Notably, innovations like DiT and U-ViT(Peebles & Xie, 2023; Bao et al., 2023) have introduced transformer-based architectures as alternatives or enhancements to the conventional U-Net. Transformer-based architectures introduce the attention mechanism into the diffusion framework, which endows the model with a more efficient capability to process global information. These advancements have significantly influenced the design of state-of-the-art image(Chen et al., 2023) and video(Gupta et al., 2024) synthesis systems, exemplified by Stable Diffusion 3.0(Esser et al., 2024a).

Many studies(Ye et al., 2025; Guo et al., 2025) have introduced or emphasized temporal information features in model training, thereby significantly enhancing the performance of the models. However, the focus is often placed on the temporal characteristics of external information, while the internal temporal features inherent to the model itself have not been sufficiently explored. Inspired by the adaLN module in DiT(Peebles & Xie, 2023), we introduce the denoising time steps within the diffusion model as indicator variables to dynamically adjust the constraint strength of different levels of information on the model at different times.

## 3 DYNAMIC ALIGNMENT AND STOCHASTIC DROPOUT STRATEGY

In this section, we will first present the formal definition of the task. Then, we will introduce the overall framework of DyA. Finally, we will provide detailed explanations of each component within the Dynamic Alignment and the Stochastic Dropout Strategy.

### 3.1 TASK DEFINITION

We provide a concise overview of flow-based and diffusion-based models from the unified viewpoint of stochastic interpolants(Albergo et al., 2023; Ma et al., 2024).

We consider $p(x)$ to be an unknown distribution of data $x \in X$. We use a model distribution to approximate $p(x)$, with a dataset drawn from $p(x)$. We define our task as learning a latent distribution $p(E(x))$ through a diffusion model, where $E$ represents a encoder from a pretrained autoencoder(Rombach et al., 2022), with $x \sim p_{data}(x)$.

We consider a continuous time-dependent process with a data $x_* \sim p(x)$ and a Gaussian noise $\epsilon \sim \mathcal{N}(0, \mathbf{I})$ on $t \in [0, T]$:

$$\mathbf{x}_t = \alpha_t \mathbf{x}_* + \beta_t \epsilon, \alpha_0 = \beta_T = 1, \alpha_T = \beta_0 = 0, \tag{1}$$

where $\alpha_t$ is a decreasing function of $t$ and $\beta_t$ is a increasing one. Considering the described process, there exists a Probability Flow Ordinary Differential Equation characterized by a velocity field.

$$\dot{\mathbf{x}}_t = \mathbf{v}(\mathbf{x}_t, t), \tag{2}$$

where the distribution of this ODE at $t$ can be seen as the marginal $p_t(x)$. Target can be sampled through the solution of Eq.(2) through existing ODE samplers starting from a random Gaussian noise $\epsilon \sim \mathcal{N}(0, \mathbf{I})$(Ma et al., 2024; Lipman et al., 2023).

This velocity $v(x, t)$ can be described through the following sum of two conditional expectations

$$\mathbf{v}(\mathbf{x}, t) = \mathbb{E}[\dot{\mathbf{x}}_t \mid \mathbf{x}] = \dot{\alpha}_t \mathbb{E}[\mathbf{x}_* \mid \mathbf{x}] + \dot{\sigma}_t \mathbb{E}[\epsilon \mid \mathbf{x}], \tag{3}$$

which can be approximated by training the model $v_\theta(x_t, t)$ to minimize the following training objective:

$$\mathcal{L}_{\text{velocity}}(\theta) := \mathbb{E}_{\mathbf{x}_*, \epsilon, t}\left[\|\mathbf{v}_\theta(\mathbf{x}_t, t) - \dot{\alpha}_t \mathbf{x}_* - \dot{\sigma}_t \epsilon\|^2\right], \tag{4}$$

Following (Ma et al., 2024), we primarily employ a straightforward linear interpolant with restricting $T = 1 : \alpha_t = 1 - t$ and $\beta_t = t$.

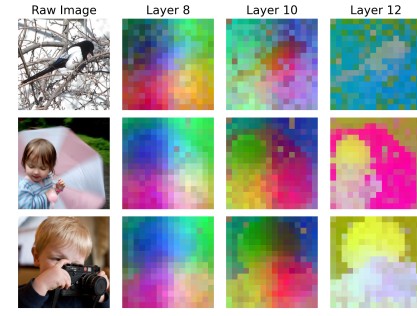

Figure 3: DyA employs visual representations at different levels to guide the training of the generation process and adjusts the guidance strength based on the denoising time as an indicator variable. This guidance is further regulated by the SDS strategy: at every training step, a Bernoulli sample decides whether to enable the DyA loss, thereby preventing over-alignment.

## 3.2 FRAMEWORK

As shown in Figure 3 DyA extracts the output of a certain layer of the diffusion model for alignment. After the output is resized by Multilayer Perceptron Layer, it is then compared with pre-trained visual representations of different level to calculate similarity and use them as Semantic Loss and Texture Loss respectively. The strength of the Texture Loss is dynamically adjusted based on the denoising time $t$ as an indicator, with its intensity varying according to the changes in denoising time.

## 3.3 DYNAMIC ALIGNMENT

Simple teacher representations for distillation can lead to capacity mismatch issues in the later stages of training. The pre-trained model provides a clear guidance for the information flow in generation tasks, but due to the inherent differences in tasks, the information contained in the generation task is far more complex than the guidance provided by the pre-trained model. After a certain stage of training, the simple guidance provided by the pre-trained model can suppress the training of generative capabilities.

Therefore, we adopt a very concise and straightforward approach to address this issue: increasing the richness of the guidance representations. We extract information from the shallow layer of the encoder to serve as the alignment target. Providing the model with more complex guidance but also alleviates the issue of over-alignment during training to a certain extent.

Figure 4: Visualization of intermediate representations of pre-trained visual encoder.(Oquab et al., 2023).

As can be seen from Figure 4, the shallow representations show a more distinct separation between objects with different semantics in adjacent areas. In contrast, deeper representations group them more roughly into one category. However, the even shallower representations are too vague to serve as a guiding factor for convergence. Such observations are consistent with the results of the following ablation experiments on the guidance layer.

Subsequently, we introduce the denoising time steps of the diffusion model as an indicator variable, dynamically adjusting the constraint strength of different levels of information on the model according to the varying denoising times. The mapping from temporal information to variables is accomplished by the **Temporal Module**, taking $t \in [0, 1]$ as input and returning a parameter $\xi \in [0, 1]$.

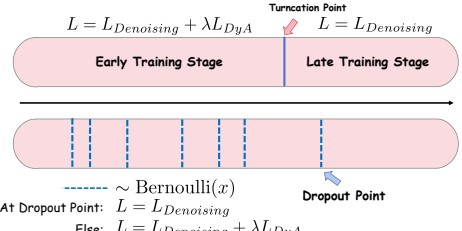

Figure 5: Visualization of SDS and Truncation.

## 3.4 STOCHASTIC DROPOUT STRATEGY

To further address the issue of model over-alignment, we propose the Stochastic Dropout Strategy (SDS), a training scheme that employs probabilistic dropout of the alignment method. To be specific, SDS will cease to align representation with a probability $p$ (where $p$ is a pre-set probability value) in a single training iteration. As

a result, the model maintains the denoising objective as the sole loss value in that iteration, while remaining unchanged in other circumstances.

Compared to truncation-based approaches, the Stochastic Dropout Strategy (SDS) not only allows the model to emphasize its generative capabilities but also provides guidance throughout the entire training process. Experimental results demonstrate that the SDS approach leads to significantly better model performance.

## 3.5 LOSS DESIGN

Considering $E$ to be a pretrained encoder and $x_*$ a clean image. Let $z_*^k = E(x_*) \in \mathbb{R}^{N \times D}$ to be the output of the encoder, where $N, D > 0$ are the number of patches and the embedding dimension of $E$ respectively, and $k$ stands for the index of the output layer of encoder. DyA aligns $h_\phi(\mathbf{h}_t) \in \mathbb{R}^{N \times D}$ with $z_*^k$ and $z_*^{final}$, where $h_\phi(\mathbf{h}_t)$ represents the intermediate output of a diffusion transformer at a specific time step $t$.

DyA achieves alignment through a maximization of patch-wise similarities between texture representation $z_*^k$, semantic representation $z_*^{final}$ and the hidden state $\mathbf{h}_t$, with $\gamma$ as a time-related variable used for adjusting the regularization intensity of texture:

$$\mathcal{L}_{\text{DyA}}(\theta, \phi, t) := -\mathbb{E}_{\mathbf{x}_*, \epsilon, t} \left[ \xi A(z_*^k, h_\phi(\mathbf{h}_t)) + A(z_*^{final}, h_\phi(\mathbf{h}_t)) \right], \tag{5}$$

where $A$ means alignment and

$$\xi = T_\sigma(t), \tag{6}$$

$T_\sigma$ is a temporal module that takes a time embedding as input and produces the variable $\xi$ via a linear transformation. The first and second alignment equal to

$$A(z_*^k, h_\phi(\mathbf{h}_t)) := \frac{1}{N} \sum_{n=1}^{N} \text{sim}(z_*^{k[n]}, h_\phi(\mathbf{h}_t^{[n]})), \tag{7}$$

$$A(z_*^{final}, h_\phi(\mathbf{h}_t)) := \frac{1}{N} \sum_{n=1}^{N} \text{sim}(z_*^{final[n]}, h_\phi(\mathbf{h}_t^{[n]})), \tag{8}$$

where $n$ is a patch index and $sim(,)$ is cosine similarity function.

We define a random variable $D \sim Bernoulli(p)$ to indicate whether to use the DyA regularization in a particular iteration, where $P(D = 1) = p$ and $P(D = 0) = 1 - p$.

We present the complete algorithm in Appendix A and ultimately obtain the DyA loss term. In practice, we add this term to the original diffusion-based objectives which give us the final loss function:

$$L := L_{Diff} + \lambda L_{DyA}, \tag{9}$$

where $\lambda > 0$ is a hyperparameter that controls the trade off between denoising object and flexible regularization intensity noted as $L_{DyA}$ in the above equation.

Table 1: Model configuration details.

| Config | Layers | HiddenDim | Heads | Student Layer | Alignment Target | Params |
|--------|--------|-----------|-------|---------------|------------------|--------|
| B/2 | 12 | 768 | 12 | L6 | L9 + L11 | 142M |
| L/2 | 24 | 1024 | 16 | L8 | L9 + L11 | 470M |
| XL/2 | 28 | 1152 | 16 | L8 | L9 + L11 | 687M |

## 4 EXPERIMENT

**Implement details**. We follow the setup in SiT(Ma et al., 2024) and REPA(Yu et al., 2025). We use ImageNet(Deng et al., 2009) and Artbench(Liao et al., 2022) to carry out our experiment, and each image is processed to resolution 256×256. We follow ADM(Dhariwal & Nichol, 2021) for data preprocessing protocols and evaluation metrics. Each image is encoded into a compressed vector $z \in \mathbb{R}^{32 \times 32 \times 4}$ using Stable Diffusion VAE(Rombach et al., 2022). Model details can be seen in Table 1. Training the L size DyA model for 400K iterations with 2 A100 GPUs and in a batch size of 256 takes about 97 hours.

Table 2: FID comparisons with vanilla SiTs, DiTs and REPA on ImageNet 256×256. We do not use classifier-free guidance (CFG). ↓ denotes lower values are better. Iter. indicates the training iteration.

| Model | FLOPs | Iter. | FID↓ |
|---|---|---|---|
| DiT-B/2 | 23.0 | 400K | 43.5 |
| **+DyA(ours)** | 23.0 | 400k | **34.7** |
| DiT-L/2 | 80.7 | 400K | 23.3 |
| +REPA | 80.7 | 400K | 15.6 |
| **+DyA(ours)** | 80.7 | 400K | **12.8** |
| SiT-B/2 | 21.8 | 400K | 33.0 |
| +REPA | 21.8 | 400K | 24.4 |
| **+DyA(ours)** | 21.8 | 400K | **22.6** |
| SiT-L/2 | 77.5 | 400K | 18.8 |
| +REPA | 77.5 | 400K | 10.0 |
| **+DyA(ours)** | 77.5 | 400K | **9.0** |
| SiT-XL/2 | 117.7 | 400K | 17.2 |
| +REPA | 117.7 | 400K | 7.9 |
| **+DyA(ours)** | 117.7 | 400K | **7.5** |

Table 3: System-level comparison on ImageNet 256×256 without CFG (classifier guidance). DyA use the 9th layer and the final layer as the alignment target and use $\lambda = 1$. The ↓ and ↑ indicate whether lower or higher values are better, respectively.

| Model | Epochs | FID↓ | IS ↑ | Rec. ↑ |
|---|---|---|---|---|
| *Pixel diffusion* | | | | |
| ADM-U | 400 | 3.94 | 186.7 | 0.52 |
| VDM++ | 560 | 2.40 | 225.3 | - |
| Simple diffusion | 800 | 2.77 | 211.8 | - |
| CDM | 2160 | 4.88 | 158.7 | - |
| *Latent Diffusion,Unet* | | | | |
| LDM-4 | 200 | 3.60 | 247.7 | 0.48 |
| *Latent Diffusion,Transformer+U-Net hybrid* | | | | |
| DiffT | - | 1.73 | 276.5 | 0.62 |
| U+ViT-H/2 | 240 | 2.29 | 263.9 | 0.57 |
| *Latent Diffusion, Transformer* | | | | |
| MaskDiT | 1600 | 2.28 | 276.6 | 0.61 |
| SD-DiT | 480 | 3.23 | - | - |
| DiT-XL/2 | 1400 | 2.27 | 278.2 | 0.57 |
| SiT-XL/2 | 1400 | 2.06 | 270.3 | 0.59 |
| +REPA | 800 | 1.80 | **284.0** | 0.61 |
| **+DyA(ours)** | 200 | **1.71** | 254.9 | **0.63** |
| **+DyA(ours)** | 400 | **1.59** | 264.0 | **0.64** |
| **+DyA(ours)** | 600 | **1.55** | 272.4 | **0.65** |

## 4.1 SETUP

**Evaluation metric**. We implement Frechet Inception Distance (Heusel et al., 2017), sFID (Nash et al., 2021), Inception Score (Salimans et al., 2016), Precision (Pre.) and Recall (Rec.)(Kynkäänniemi et al., 2019) on 50,000 samples.

**Sampler**. We follow SiT(Ma et al., 2024) and use the SDE Euler-Maruyama sampler (for SDE with $\omega_t = \sigma_t$) and set the number of function evaluations (NFE) as 250 by default.

**Baselines**. We take several recent diffusion-based generation methods into consideration, each employing different inputs and network architectures. Four types of approaches are included: **Pixel diffusion**, **Latent diffusion with U-Net**, **Latent diffusion with transformer+U-Net hybrid models** and **Latent diffusion with transformers**.

## 4.2 SYSTEM-LEVEL COMPARISON

We conduct a systematic comparison between recent state-of-the-art diffusion model approaches, diffusion transformers with REPA, and diffusion transformers with DyA. First, we compare the FID values between vanilla SiT, vanilla DiT, SiT with REPA, DiT with REPA and the same models trained with DyA. As shown in Table 2, DyA achieves comprehensive improvements over vanilla SiT, vanilla DiT and REPA. Specifically, SiT achieves FID of 17.2

Table 4: Comparison on Artbench 256×256 with no CFG. The ↓ and ↑ indicate whether lower or higher values are better, respectively.

| *SiT* | sFID↓ | Pre.↑ | Rec.↑ |
|---|---|---|---|
| +REPA | 29.68 | 0.55 | 0.35 |
| +DyA | 31.70 | 0.54 | 0.35 |
| +DyA with Truncate | 27.48 | **0.61** | 0.31 |
| +DyA with SDS | **26.67** | 0.60 | **0.36** |

at 400k iterations and with the help of REPA achieves FID of 7.9 at 400k iterations, while DyA reaches FID of 7.5 at same iterations. The model trained with DyA exhibits even better convergence speed than REPA. The experimental results are averaged over five independent runs. A Wilcoxon rank-sum test reveals that DyA significantly outperforms REPA in both FID and IS ($p < 0.05$, Rank-Biserial $r = 0.6$), indicating a statistically meaningful difference.

We provide a quantitative evaluation comparing SiT-XL/2 with DyA to all model variants. As result can be seen from Table 3, DyA shows consistent and significant improvement. At 200 epochs, SiT-XL/2 with DyA achieves FID of 1.71 with a classifier-free guidance scale of $\omega = 1.5$ , already better than all previous methods. Our method outperforms the original SiT-XL/2 with 7× fewer epochs and out performs REPA with 4× fewer epochs and it is further improved with longer training. As the number of model training epochs increases, the FID shows a continuous decline, further dropping to 1.59 at the 400 epochs. The Rec. value reached 0.65 at 600 epoch which is better than all previous

Table 5: **Component-wise analysis** on ImageNet 256×256. All models are SiT-L/2 trained for 400K iterations. All metrics are measured with the SDE Euler-Maruyama sampler with NFE=250. We fix $\lambda = 1$ here. ↓ and ↑ indicate whether lower or higher values are better, respectively.

| Iter. | Time Modulate | Guidance | Dropout Rate | FID↓ | sFID↓ | IS↑ | Pre.↑ | Rec.↑ |
|---|---|---|---|---|---|---|---|---|
| 400K | - | L11 | - | 10.0 | 5.34 | 111.9 | 0.68 | 0.65 |
| 400K | $1 - \xi, \xi$ | +L7 | - | 10.4 | 5.20 | 106.9 | 0.68 | 0.65 |
| 400K | $\xi$ | +L7 | - | 9.39 | 5.09 | 114.1 | 0.69 | 0.65 |
| 400K | $\xi$ | **+L5** | - | 9.65 | 5.15 | 112.2 | 0.68 | 0.65 |
| 400K | $\xi$ | **+L7** | - | 9.39 | 5.09 | 114.1 | 0.69 | 0.65 |
| 400K | $\xi$ | **+L9** | - | 9.00 | 5.17 | 116.1 | 0.69 | 0.65 |
| 400K | $\xi$ | **+L9** | 5% | 9.23 | 5.19 | 114.8 | 0.69 | 0.65 |
| 400K | $\xi$ | **+L9** | 10% | 9.18 | 5.28 | 116.1 | 0.68 | 0.66 |
| 400K | $\xi$ | **+L9** | 15% | 9.19 | 5.31 | 115.8 | 0.68 | 0.65 |
| 400K | $\xi$ | **+L9** | 20% | 9.08 | 5.17 | 116.0 | 0.69 | 0.65 |
| 400K | $\xi$ | **+L9** | 25% | 8.96 | 5.56 | 117.4 | 0.70 | 0.65 |
| 400K | $\xi$ | **+L9** | Truncation | 17.44 | 6.66 | 80.7 | 0.58 | 0.68 |

method listed above. We can clearly see from the table that both the FID and IS scores improve significantly with the increase in training epochs.

To investigate the generalization capability of our model and its hyper-parameter choices across diverse datasets, we conduct training and evaluation on ArtBench. Training was performed for $50\,000$ steps with a batch size of $512$, using SiT-B/2 as the backbone and comparing both REPA and the truncation training strategy. Our results demonstrate that DyA combined with the SDS training schedule achieves competitive performance.

## 4.3 ABLATION STUDY

We have experimented with two different ways of using the **Temporal Module**. In our initial attempt, we use $\xi$, the output of the temporal module, to adjust the regularization intensity of both semantic and texture information, where $\xi \in [0, 1]$. We set $\xi$ and $1 - \xi$ as the regularization intensities for the two types of information, respectively. As shown in Table 5, although the model's performance improved in terms of sFID, its performance in IS and FID was lower than that of REPA. This indicates that the regularization intensity demands for different types of information during the denoising process are not simply inversely proportional. In the subsequent experiments, we adopt the latter approach.

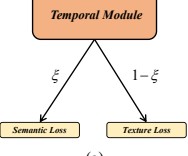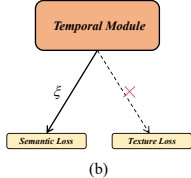

Figure 6: (a) represents the method in which temporal module regulate both semantic loss and texture loss. (b) shows the method that temporal module regulate texture loss only.

Then we conducted ablation studies on different **Guidance**. As result can be seen from Table 5, DyA with the modification outperforms REPA in all five metrics, with the improvement being particularly evident in the sFID score. From the data, we can observe that, at a fixed number of training steps, increasing the richness of guidance can significantly enhance the training efficiency of generative models. Additionally, as visualized in Figure 4, features that emerge too early are not suitable for use as generative guidance. A plausible explanation is that the image information has not been processed through a sufficient number of layers to produce convergent guiding representations.

From Table 5 it can be discovered that DyA achieves better evaluation score regardless of the layer choice of alignment target, and we attribute this phenomenon to the presence of temporal module. Additionally, the performance gains contributed by different layers are non-uniform. Therefore To preserve methodological simplicity and uphold the algorithmic essence–enhancing guidance complexity in the most straightforward manner–we ultimately employ double target layers.

We have evaluated the model with the **Stochastic Dropout Strategy** in different dropout rate. As shown in Table 5, we experiment with dropout rates of 5%, 10%, 15% , 20% and 25% for the L-sized model. Overall, we find that DyA still outperforms baseline. Moreover, compared with truncation strategy which ceases the alignment from a pre-set step, SDS also have a better performance. Com-

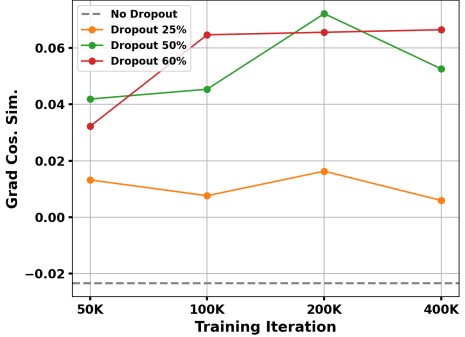
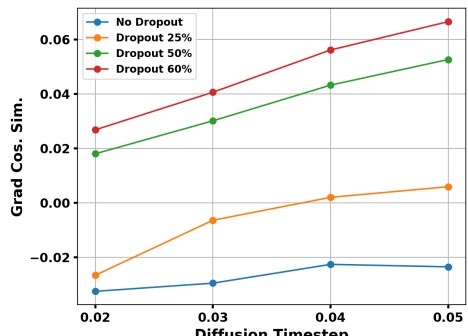

Figure 7: The impact of the SDS strategy on the generation effect. A high dropout rate can lead to insufficient pre-training guidance strength, thereby slowing down the convergence speed. Furthermore, the truncation method generally yields inferior generation quality before the truncation point compared to the SDS approach.

Figure 8: Gradient angle between Denoising Loss and Auxiliary Loss across training iteration.

Figure 9: Gradient angle between Denoising Loss and Auxiliary Loss across denoising timesteps.

pared to the truncation approach, the guidance provided by SDS spans the entire training process. It also does not significantly affect the generation quality due to the setting of a too early or too late turncation point.

During training, we observed that the diffusion loss of the SDS-augmented model is consistently higher than its REPA counterpart in the early phase. Once the training step exceeds 30 000, the auxiliary loss decreases at a significantly faster rate. Such a situation indicates that, from a training dynamic perspective, the impact of SDS on the generative-related denoising loss is superior to full-process alignment, effectively alleviating the issue of over-alignment.

Furthermore, from Figure 7 the truncation method generally yields inferior generation quality before the truncation point compared to the SDS approach. This indicates that issues of over-alignment and capacity mismatch do not solely emerge in the later stages of training.

Thus, it can be concluded that the underlying assumption of truncation, which posits that over-alignment and capacity mismatch occur only in the later stages of training, is erroneous. In contrast, the SDS strategy does not rely on such a premise. Instead, the SDS approach attributes the problem to the inherent differences in the tasks of various models, adjusting guidance throughout the entire training process. This effectively enhances both the quality of generation and the efficiency of training.

From a gradient perspective, as can be seen from Figure 9 and Figure 8, A higher dropout rate has a more pronounced effect in mitigating the issue of over-alignment. At dropout rates of 50% and 60%, the gradients of the auxiliary loss and the denoising loss remain positive throughout the entire training process, indicating that the guidance consistently contributes to positive optimization of the generation effect. However, from the perspective of the generation effect, the positive cosine similarity is actually due to "insufficient alignment." As can be seen from Figure 7, the images with a 25% dropout rate clearly show more mature image generation at earlier training steps.

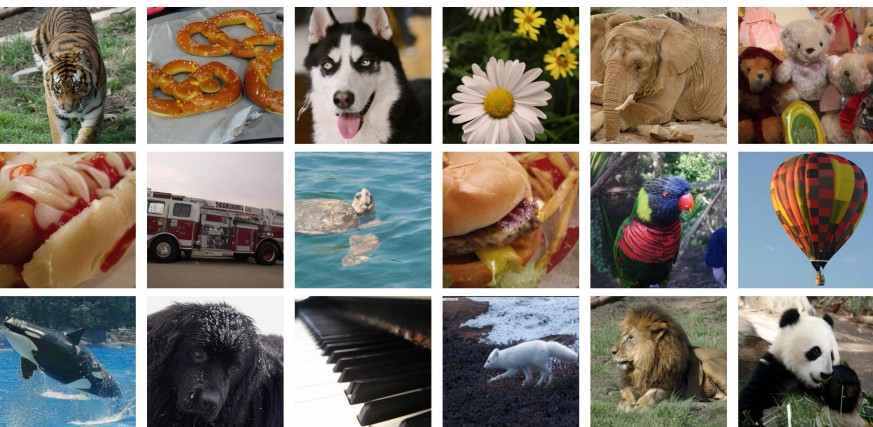

Figure 10: Selected samples from models trained on ImageNet 256 ×256 resolution with cfg = 4.0.

Using the SDS strategy requires a balance between the positive optimization brought by alignment and the actual generation results. Through experimentation, we found that a dropout rate of 25% provides the greatest enhancement to generation quality, while also ensuring that the alignment strategy continues to provide positive guidance for the generation capability training throughout the long-term training process.

From Figure 9, we can see SDS actually alleviate the over-alignment issue. From denoising time 0.02 to 0.05, the cosine similarity between gradient of the auxiliary loss and the gradient of the denoising loss are negative without the use of the SDS strategy. With the implementation of the SDS strategy, the cosine similarity during this time period is almost entirely positive. As mentioned above, considering the dual aspects of balancing guidance for positive optimization and generation quality, we ultimately adopted a dropout rate of 25% as the final solution.

Table 6: Ablation study for Stu. Layer .

| Stu. Layer | Layer6 | Layer8 | Layer10 |
|---|---|---|---|
| FID↓ | 22.6 | 26.4 | 34.4 |
| IS↑ | 65.44 | 57.91 | 46.30 |

Table 7: Ablation study for $\lambda$.

| $\lambda$ | 0.25 | 0.5 | 0.75 | 1 |
|---|---|---|---|---|
| FID↓ | 10.21 | 10.10 | 9.79 | **9.39** |
| IS↑ | 107.1 | 108.8 | 111.5 | **114.1** |

We investigate the impact of different **Student Layers** on model performance. As shown in Table 6, we find that the model's performance actually shows a decline as the relative depth of the alignment layer increases, consistent with the findings in the REPA.

We examine the effect of the regularization coefficient $\lambda$ by training DyA with different coefficients 0.5 to 1.0 and comparing the FID and IS. As shown in Table 7, the performance have stability over different values and peak at $\lambda = 1$. Although performance exhibits a monotonic rise with increasing $\lambda$, we ultimately cap $\lambda$ at 1 to keep the optimization focus unambiguously on generation quality.

## 5 CONCLUSION

In this paper, we presented Dynamic Alignment. We introduced multi-level information coupled with temporal module to address the capacity mismatch issue in conventional representation-guided generation. Building upon this, Stochastic Dropout Strategy was further employed to mitigate over-alignment issue. Systematic comparisons on ImageNet and ArtBench demonstrated that DyA significantly enhances the generation quality of model and accelerate training convergence.

We empirically observed that augmenting representation richness together with the SDS strategy consistently elevates generation quality. And semantic structures are able to emerge markedly earlier in the target layer. These findings corroborate the validity of our methodological framework. We will endeavor to transcend the constraints of pre-trained models and further refine this paradigm.

ETHICS STATEMENT

Our work investigates diffusion models for generative modeling. We believe this research has the potential to contribute positively by enabling creative applications, advancing understanding of probabilistic modeling, and improving efficiency in downstream scientific and industrial tasks.

However, we acknowledge possible negative impacts. The proposed method could be misused for generating misleading, harmful, or otherwise inappropriate content. Our experiments are conducted exclusively on publicly available benchmark datasets, which do not contain personally identifiable information. Nevertheless, any biases present in the datasets may be propagated or amplified by our models. We encourage further work on bias detection and mitigation in generative modeling.

Regarding environmental impact, our experiments were performed on $4 \times$ A100 GPUs. We recognize the importance of efficient model design and responsible use of computational resources in order to reduce the carbon footprint of large-scale model training.

Overall, we emphasize that our research should be used for beneficial purposes only, and we discourage applications that may cause harm to individuals or society.

REPRODUCIBILITY STATEMENT

We are committed to ensuring the reproducibility of our work. All datasets used in our experiments are publicly available. We will release our code, along with detailed instructions for training and evaluation.

We describe all necessary implementation details in the main papersupplementary material, and appendix, including model architectures, optimization settings (learning rate, batch size, optimizer, scheduler), and data preprocessing steps. We set fixed random seeds for all experiments to ensure consistent results across runs.

Our experiments were conducted on $4 \times$ A100 GPUs.

All reported results in tables and figures can be reproduced using the released code and configuration files. We also provide scripts to regenerate the main figures and evaluation metrics directly from trained checkpoints.

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

## A  ALGORITHM

---

**Algorithm 1** DyA with SDS.

---

**Input:** probability, batchsize, representations, timestep and target;
**Output:** $L_{DyA}$;
  $semantic, texture = target$;
  $Dropout = random() > probablity$;
  $L_{DyA} = 0$;
  **if** not $Dropout$ **then**
    $\xi = TemporalModule(timestep)$
    **for** $representation$ in $representations$: **do**
      $loss_{sem} = 0, loss_{tex} = 0$;
      **for** $batch$ in $representation$ **do**
        $loss_{sem} + = sim(batch, semantic)$;
        $loss_{tex} + = sim(batch, texture)$;
      $L_{DyA} + = (loss_{sem} + \xi * loss_{tex})/batchsize$;
    **end for**
  **end if**
  **return** $L_{DyA}$

---

## B  ACKNOWLEDGEMENTS

We used large language models (LLMs) such as ChatGPT for minor editing support, including grammar checking and language polishing. No LLMs were used for generating technical content, experimental results, or ideas in this work.

