# OpenReview forum: "Accelerating Diffusion Model with Dynamic Alignment"
_ICLR.cc/2026/Conference — Submitted to ICLR 2026_

### Official Review · Reviewer_6FJa · 2025-10-30

**Soundness:** 3
**Presentation:** 2
**Contribution:** 3
**Rating:** 4
**Confidence:** 4

**Summary:**

The paper presents Dynamic Alignment (DyA), a multi-level guidance method (semantic and texture) that dynamically adjusts alignment strength across timesteps using a temporal module for training diffusion transformers. It also introduces a Stochastic Dropout Strategy (SDS) that randomly disables alignment loss to better preserve generative capacity. Experiments show that DyA accelerates training compared to naive external alignment while improving generation quality.

**Strengths:**

- The motivation of the paper is clear - capacity mismatch problem in naive representation alignment.

- Experiments demonstrate consistent improvement over REPA (naive representation alignment) and ablation studies show that both DyA and SDS contribute meaningfully to the gains.

**Weaknesses:**

- While the paper argues that using representation from different layers of the encoder (semantic vs. texture) provides richer guidance, it doesn’t explain what those signals actually contain or how they help the diffusion transformer training. More analysis is needed to show whether this observation holds across different pretrained encoders (e.g., DINOv2, CLIP) and whether the method works consistently across multiple encoders.

- The temporal module determines how much weight to give to semantic and texture losses at different timestep, but the paper doesn’t show how this weighting changes (e.g., are semantic cues more important at small timestep?).

- Table 3 is labeled “no CFG,” yet the text (Line 320) reports results with $\omega$=1.5. This needs to be clarified.

- Implementation details are missing - which encoder is used, and the exact architecture of the temporal module. Brief descriptions (with citations) of baselines would improve clarity and reproducibility.

**Questions:**

- The overall writing could be improved for better readability. In particular, Section 3.1 is very similar to that in REPA - it would be better if the authors provided their own explanation, even though this part is standard background. The related work section should also be reorganized to more clearly position this paper, perhaps including recent works on efficient diffusion transformer training via alignment.
- If Table 3 reports results without CFG, I am curious about results when CFG is applied.
- Can the proposed method be easily extended to text-to-image generation tasks?

---

### Official Review · Reviewer_sCk2 · 2025-10-31

**Soundness:** 3
**Presentation:** 2
**Contribution:** 2
**Rating:** 4
**Confidence:** 3

**Summary:**

This paper proposes a training framework for diffusion models called Dynamic Alignment (DyA), which combines multi-level feature alignment with a temporally adaptive modulation mechanism. The method addresses representation-guided generation’s capacity mismatch from both semantic and texture perspectives. In addition, a Stochastic Dropout Strategy (SDS) is introduced to probabilistically control alignment strength, mitigating over-alignment in later training stages. Experiments on ImageNet and ArtBench show that the proposed method achieves superior generation quality and training efficiency compared with existing baselines.

**Strengths:**

1. **Clear problem identification and targeted solution:** DyA introduces temporal information into the alignment modulation process, forming a multi-level dynamic guidance mechanism that effectively alleviates the capacity mismatch problem in representation-guided generation.
2. **Improved stability and efficiency:** The SDS strategy offers a probabilistic and flexible alignment scheme, avoiding the instability of truncation-based distillation while achieving faster convergence and better generative quality.
3. **Comprehensive empirical validation:** The paper presents detailed comparisons and ablation studies on both ImageNet and ArtBench, convincingly demonstrating the effectiveness of the proposed approach.
4. **Simple and extensible design:** DyA is architecture-agnostic and, in principle, can be integrated into various diffusion frameworks. This generality is appealing, although no experiments currently verify this claim.

**Weaknesses:**

1. **Empirical rather than theoretical depth:** Despite solid experiments, the contribution appears more like an engineering-oriented extension of REPA, with limited theoretical novelty.
2. **Inconsistent presentation of results:** The tables are presented inconsistently — e.g., Table 2 uses *Iter.*, Table 3 uses *Epochs*; some tables report FID only, while others include IS, Rec., or sFID. Only Table 5 reports all metrics together. This inconsistency makes it difficult to clearly interpret the comparative performance across experiments. A unified reporting format (e.g., using the same metrics across tables) would greatly improve readability and credibility.

**Questions:**

1. For *Fig. 2* (and related descriptions), is it a conceptual illustration (hypothesis) or derived from empirical results? It seems inconsistent with the gray dashed “No Dropout” curve in Fig. 8, which remains below zero.
2. In *Table 3*, why is DyA not trained for the same epochs as REPA? A fair comparison would require identical training durations.
3. The performance degradation observed with *Truncation* in Table 5 (compared to the no-Dropout, no-Truncation case) appears contradictory to the results in Table 4 and to the overall intuition of the paper. Could the authors clarify this discrepancy?
4. Could you explicitly show the mapping curve between $t$ and $\xi$ in the Temporal Module after training? A figure depicting this relationship would provide stronger empirical evidence supporting the paper’s conceptual claims.

---

### Official Review · Reviewer_8zLX · 2025-10-31

**Soundness:** 2
**Presentation:** 1
**Contribution:** 3
**Rating:** 2
**Confidence:** 4

**Summary:**

This paper aims to accelerate the training of diffusion transformers by aligning pre-trained representations but reducing over-alignment. The authors first observe that overly aligned representations hinder the training of diffusion models for generation. To address this, they propose DyA that uses richer alignment materials (semantic + texture) and dynamically adjusts the alignment objective. They demonstrate that DyA accelerates the training of the diffusion model faster than REPA.

**Strengths:**

1. The paper is well-motivated in that over-alignment may hinder learning the model to generate data.

2. The proposed method (DyA) shows better performance than REPA consistently.

**Weaknesses:**

1. Missing analysis on $\xi$: What is the reason that uses learnable parameter $T_\sigma$ to obtain $\xi$? Comparison with linear transformations (e.g., $\xi=t$ or $\xi=1-x$) or well-known transformations (e.g., $\xi=\text{sigmoid}(t)$) is needed. This is important to understand how $\xi$ affects the training of diffusion transformers.

2. Incorrect component-wise analysis, which makes it difficult to know the effectiveness of the methodology: In Dropout Rate analysis, 25% is the best, but other analysis is conducted without dropout rate (9.00, which is not the best score). In addition, L9 guidance shows the best among the options, but analysis on time modulation and baseline is conducted using L11 guidance. To demonstrate the effectiveness of each component, an ablation study is needed.


**Minor**
1. In line 238, I think $T_\sigma$ is not a linear transformation (see Fig 3, which includes SiLU activations.)

2. In Figure 6, semantic loss seems to be regularized. Moreover, I think the reference to Figure 6 is incorrect.

**Questions:**

1. Please answer the Weakness.

2. What is texture representation? It is difficult to understand how the texture representations are obtained. Are the intermediate features called texture? Then, why should we call the intermediate features texture? For instance, texture may be obtained by applying some filters (using convolution operations).

---

### Official Review · Reviewer_SsgG · 2025-10-31

**Soundness:** 3
**Presentation:** 3
**Contribution:** 2
**Rating:** 2
**Confidence:** 4

**Summary:**

This paper aims to improve representation alignment-based diffusion model training. Prior "representation alignment" speeds up diffusion transformer training, but the paper argues that the teacher's features can be too simple and cause over-alignment, hurting generative capability. The author tries to resolve this via DyA, which is equipped with 1) richer targets, 2) Time-aware strength, and 3) SDS (Stochastic Dropout Strategy). For richer targets, Dya utilizes multiple intermediate features for a shallow layer and the later layer, avoiding that guidance is not overly simplified. Then, alignment strength is modulated with Time-aware strength and stochastically dropped according to a defined probability.

**Strengths:**

- Well-motivated upgrade over REPA. Clearly identifies and fixes over-alignment via time-aware, multi-level feature guidance and stochastic dropout.
- Strong empirical gains. Consistently better FID/IS and recall than REPA/SiT/DiT baselines at matched steps.

**Weaknesses:**

- This work completely ignores previous acceleration methods for diffusion model training. Even if they improve the REPA, these previous acceleration works should be discussed and compared. Since [1] shows REPA is actually comparable to or underperforms previous acceleration methods, the author should compare their approach with the following works:
  - A Closer Look at Time Steps is Worthy of Triple Speed-Up for Diffusion Model Training, CVPR 2025. (Timestep sampling + loss weighting)
  - Efficient Diffusion Training via Min-SNR Weighting Strategy, ICCV 2023.  (loss weighting)
  - Addressing Negative Transfer in Diffusion Models, Neurips 2023. (loss weighting)
  - Perception prioritized training of diffusion models, CVPR 2023. (loss weighting)
  - Fast training of diffusion models with masked transformers, TMLR 2024. (masked transformer)
  - Beta-tuned timestep diffusion model, ECCV 2024. (timestep sampling)
  - Denoising task difficulty-based curriculum for training diffusion models, ICLR 2025. (curriculum learning)
  - Non-uniform timestep sampling: Towards faster diffusion model training, ACMMM 2024 (timestep sampling)
- If this work outperforms or has a clear advantage for faster convergence compared to the above methods, I will lean towards acceptance. Otherwise, I think that the contribution of this work is not enough.

Reference

[1] Bidirectional Beta-Tuned Diffusion Model, TPAMI 2025.

**Questions:**

N/A

---

### Meta-Review · Area_Chair_xLSS · 2026-01-07

**Summary:**

The paper received uniformly negative reviews, with no discussion or author response available. Based on the reviewers’ feedback, the area chair recommends rejection.

**Reviewer Concerns:**

all of the concerns are still outstanding

**Reviewer Scores:**

n/a

---

### Decision · Program_Chairs · 2026-01-26

Reject